# Molecular Biomarkers and Therapeutic Approach of Patients with Diabetes and Obstructive Sleep Apnea

**DOI:** 10.3390/ijms262010234

**Published:** 2025-10-21

**Authors:** Viviana Elian, Violeta Popovici, Alexandru Tudor Steriade, Gabriela Radulian, Emma Adriana Ozon, Elena Moroșan, Madalina Musat

**Affiliations:** 1Diabetes, Nutrition and Metabolic Disease Unit, Department of Diabetes, “Carol Davila” University of Medicine and Pharmacy, 020475 Bucharest, Romania; viviana.elian@umfcd.ro (V.E.); gabriela.radulian@umfcd.ro (G.R.); 2Diabetes, Nutrition and Metabolic Disease Unit, National Institute of Diabetes, Nutrition and Metabolic Disease, Prof. N. C. Paulescu, 020475 Bucharest, Romania; 3Center for Mountain Economics, “Costin C. Kiritescu” National Institute of Economic Research (INCE-CEMONT), Romanian Academy, 725700 Vatra-Dornei, Romania; 4Department of Cardio-Thoracic Pathology, “Carol Davila” University of Medicine and Pharmacy, 020021 Bucharest, Romania; alexandru.steriade@umfcd.ro; 5Department of Pneumology & Acute Respiratory Care, “Elias” Emergency University Hospital, 011461 Bucharest, Romania; 6Department of Pharmaceutical Technology and Biopharmacy, Faculty of Pharmacy, “Carol Davila” University of Medicine and Pharmacy, 020945 Bucharest, Romania; emma.budura@umfcd.ro; 7Department of Clinical Laboratory and Food Safety, Faculty of Pharmacy, “Carol Davila” University of Medicine and Pharmacy, 020945 Bucharest, Romania; elena.morosan@umfcd.ro; 8Department of Endocrinology, “Carol Davila” University of Medicine and Pharmacy, 020021 Bucharest, Romania; 9Department of Endocrinology, C. I. Parhon National Institute of Endocrinology, 011683 Bucharest, Romania

**Keywords:** obstructive sleep apnea, type 2 diabetes, molecular markers, molecular mechanisms, GLP-1RAs, tirzepatide, SGLT2 inhibitors, clinical evidence

## Abstract

The bidirectional relationship between obstructive sleep apnea (OSA) and type 2 diabetes mellitus (T2DM) represents a critical intersection in metabolic medicine. Therefore, the present review examines the most recent data regarding molecular mechanisms linking OSA and T2DM, analyzing key biomarkers including hypoxia-inducible factors (HIF 1α), inflammatory mediators, adipokines, microRNAs, hormones, and neuropeptides that serve as both diagnostic indicators and potential therapeutic targets. Key molecular findings from the scientific literature report elevated HIF-1α promoting insulin resistance, decreased SIRT1 levels, dysregulated microRNA-181a and microRNA-199a, increased inflammatory cytokines (TNF-α, IL-6, CRP), and altered adipokine profiles with reduced adiponectin and elevated leptin and resistin. Current clinical evidence reveals significant therapeutic potential for modern antidiabetic medications in the management of OSA. GLP-1 receptor agonists, particularly tirzepatide, received FDA approval as the first medication for moderate-to-severe OSA in obese adults, showing a 55–63% AHI reduction. SGLT2 inhibitors also demonstrate promising results through weight loss and cardiovascular protection mechanisms. This integrated approach represents the evolution toward comprehensive OSA management beyond traditional mechanical ventilation strategies. Future research should focus on developing personalized treatment algorithms based on individual molecular biomarker profiles, investigating combination therapies, and exploring novel targets, including chronotherapy agents.

## 1. Introduction

Obstructive sleep apnea (OSA) is a disorder characterized by the repetitive collapse of the upper airway during sleep, resulting in a complete or partial obstruction of the upper airway, respiratory effort-related arousals, increased oxidative stress, inflammation, sleep fragmentation, and sympathetic nervous system activation [1]. It is a chronic, treatable sleep disorder and a frequent comorbidity in patients with type 2 diabetes mellitus (T2DM) [2,3].

Recognition of OSA as a systemic disease with cardiovascular, metabolic, and neurological consequences has led to a transformation in clinical approaches [4,5,6]. It highlights the importance of comprehensive management strategies that address both the sleep disorder and its wide-ranging health implications. The co-occurrence of OSA and both type 1 and type 2 diabetes mellitus is remarkably high. OSA is frequently present in patients with T2DM, with a reported prevalence of approximately 55 to 85% among various longitudinal cohort studies [7,8]. A systematic review and meta-analysis found that the overall prevalence of OSA in patients with T2DM was 56.0%, with prevalence rates increasing with age, male sex, and higher body mass index [9]. The association between OSA and type 1 diabetes (T1DM) has also been evidenced [10,11,12,13], with an almost 51.9% [14].

The relationship between OSA and diabetes is complex and bidirectional, meaning that each condition can influence and exacerbate the other. This bidirectional relationship represents one of the most significant and complex interactions in modern sleep medicine and endocrinology [15,16]. Understanding this connection is crucial for clinicians, researchers, and patients alike.

Cardinal features of OSA, including intermittent hypoxemia (IH) and sleep fragmentation, have been linked to abnormal glucose metabolism in laboratory-based studies [8,17,18]. Intermittent hypoxemia triggers multiple pathways that are potentially involved in the pathogenesis of T2DM, including obesity, alterations in glucose metabolism, sympathetic activation, systemic inflammation, oxidative stress, liver damage, and pancreatic beta-cell dysfunction (Figure 1) [19,20,21,22].

Inversely, almost 78% of patients with T2DM had OSA, with those having uncontrolled diabetes showing significantly higher OSA prevalence (86.8%) compared to those with controlled diabetes (61.9%) [23]. Type 2 diabetes can lead to OSA development through the effects of insulin resistance and autonomic dysfunction on upper airway stability [24]. Diabetic neuropathy can affect the muscles that control breathing and upper airway tone, increasing the likelihood of airway collapse during sleep [25]. Diabetes-related weight gain, particularly associated with specific diabetes treatments such as sulfonylureas, thiazolidinediones, or insulin, can exacerbate OSA by increasing upper airway tissue mass and reducing airway diameter. Additionally, diabetes can cause fluid retention, leading to swelling in the upper airway tissues that narrow the breathing passage (Figure 1).

Numerous molecular markers and complex cellular and molecular mechanisms are involved in the bidirectional relationship between OSA and diabetes mellitus. Therefore, this review aims to analyze the most recent data on the molecular mechanisms responsible for the interdependence between OSA and T2DM. Various molecular markers can serve as target molecules for the current medication of diabetes patients with OSA; thus, the present study provides a detailed presentation of antidiabetic drugs used for this purpose and their underlying molecular mechanisms.

## 2. Molecular Markers of OSA and T2DM

The underlying significant association mechanisms between OSA and T2DM may include intermittent hypoxemia and increased oxidative stress (Figure 1), involving numerous molecular markers (Figure 2).

### 2.1. Hypoxia-Inducible Factor-1α (HIF-1α)

Hypoxia-inducible factor-1α (HIF-1α), a transcription factor essential for the cell’s adaptive responses to hypoxia, serves as a central molecular mediator linking OSA to the development of diabetes mellitus. HIF-1α plays a key role in oxygen metabolism and is upregulated in patients with OSA (Figure 2).

HIF-1α is a key regulator of metabolic processes and directly contributes to the development of insulin resistance (IR) and diabetes mellitus. Research using animal models clearly shows that a single night of intermittent hypoxia can raise fasting glucose levels [26]. In patients with OSA, HIF-1α is consistently upregulated, indicating its significant role in the onset of metabolic complications within this group. Both hyperglycemia and hypoxia are essential in the pathophysiology of diabetes-related complications, often resulting from tissues failing to respond adequately to low oxygen levels. Moreover, hyperglycemia actively inhibits the hypoxia-induced stabilization of HIF-1α protein, preventing its degradation and highlighting that mechanisms beyond proline hydroxylation are involved [27]. Importantly, studies have demonstrated that serum HIF-1α levels are markedly elevated in patients with T2DM [28].

HIF is a heterodimer composed of two units: an α-subunit, which is oxygen-regulated, and a constitutively expressed β-subunit, belonging to the helix-loop-helix Per/Arnt/Sim transcription factor family [29]. Three analogs of HIF α-subunits are known: HIF-1α, HIF-2α (established regulatory factors), and HIF-3α (with an uncertain role) [30]. The first one, HIF-1α, is the best-examined HIF α-subunit [31]. Although its transcriptional level remains stable, the HIF-1α protein is highly unstable under normoxic conditions, which is attributed to the presence of an oxygen-dependent degradation domain [31]. In low oxygen conditions (hypoxia), this hydroxylation is inhibited, stabilizing HIF-1α and allowing it to activate genes that promote survival.

HIF-1α has a low half-life time under normoxic conditions; it has two key proline hydroxylation sites, Pro402 and Pro564, which are both involved in targeting the protein for degradation. Prolyl hydroxylases (PHDs) add hydroxyl groups to specific proline residues on the HIF-1α protein. This hydroxylation creates a recognition site for the von Hippel-Lindau (VHL) tumor suppressor protein. The VHL protein is part of an E3 ubiquitin ligase complex that “marks” HIF-1α for degradation through the ubiquitin-proteasome pathway [32]. Therefore, HIF-1α levels are maintained low, and its activity is suppressed in the presence of sufficient oxygen.

The lack of oxygen inhibits PHDs, and the VHL complex cannot bind to HIF-1α. Therefore, HIF-1α is not marked for degradation and becomes stable. The stable HIF-1α can bind to its beta subunit and act as a transcription factor to turn on genes needed for adaptation to low oxygen. Upon post-translational stabilization under hypoxic conditions, the active dimeric protein complex travels to the nucleus, where it binds to hypoxia-response elements in gene promoters, significantly influencing the expression of over 100 genes [33]. In contrast, under hypoxic conditions, HIF-1α plays a pivotal role in reprogramming metabolic pathways to adapt effectively to this challenging environment [34,35,36]. Its impact on glucose metabolism is particularly significant, directing key processes such as glucose uptake, glycolysis, and the regulation of the tricarboxylic acid cycle (TCA). This dynamic interaction highlights the essential role of HIF-1α in managing the cellular response to low oxygen levels.

It mediates the regulation of numerous genes that affect these processes, which HIF-1α regulates. The majority of genes are related to glycolysis enzymes: hexokinases (HK 1,2), which transform glucose into glucose-6-phosphate; liver type phosphofructokinase (PFKL); 6-phosphofructo-2-kinase/fructose-2,6-bisphosphate-3 (PFBFK3); phosphoglycerate kinase 1 (PGK1); aldolases A and C (ALDA and ALDC); glyceraldehyde-3-phosphate dehydrogenase (GAPDH); enolase 1 (ENO1), which transforms 2-phosphoglycerate into phosphoenolpyruvate; and pyruvate kinase M (PKM), which is responsible for the final step of glycolysis. Other genes regulate (i) glucose transport, including glucose transporters (GLUT 1, 3, 4), (ii) lactate metabolism, involving lactate dehydrogenase A (LDHA) and pyruvate dehydrogenase kinase (PDK1), and (iii) maintaining pH levels during hypoxia through carbonic anhydrase 9 (CA9). Several genes are associated with other functions: metabolic regulation (thioredoxin-interacting protein, TXNIP) and vesicular trafficking (small GTP-ase, RAB20). All are presented in Figure 3.

The relationship between HIF-1α and glucose metabolism is complex. HIF-1α-regulated genes play a crucial role in glucose metabolism (Figure 3). Cell culture studies indicate that HIF-1α regulates both glucose uptake and glycolytic enzyme activity, significantly enhancing the glycolysis process as a mediator of insulin resistance. Considering that animal studies suggest HIF-1α as a potential therapeutic target in impaired glucose metabolism, this may be a promising research direction in patients with OSA [45].

The influence of HIF-1α on GLUT-4 mirrors its impact on GLUT-1, leading to a significant enhancement in glucose uptake [46]. When HIF-1α is knocked down, a marked decline in insulin-stimulated glucose uptake occurs in cultured skeletal muscle cells, primarily due to the impaired translocation of GLUT-4 to the plasma membrane [47]. While numerous studies affirm that hypoxia, coupled with HIF-1α overexpression, can detrimentally affect metabolism, some research suggests that the stabilization of HIF-1α may also yield beneficial effects on glucose and lipid metabolism [48]. In a study by Görgens et al., focusing on human skeletal muscle cells, the researchers discovered that hypoxia, in conjunction with muscle activity, significantly improves glucose metabolism and enhances insulin sensitivity through the HIF-1α pathway. This pathway exerts a powerful influence on the transcription of RAB20 and TXNIP. Rab20, part of the Rab family of proteins, plays a substantial role in regulating intracellular trafficking and vesicle formation [39]. Its deletion disrupts insulin-stimulated glucose uptake by obstructing the translocation of GLUT-4 to the cell surface. TXNIP encodes a thioredoxin-binding protein from the alpha-arrestin protein family, which serves a variety of essential functions, including the regulation of cellular metabolism [49]. Research has shown that TXNIP can enhance insulin secretion and modulate glucagon-like peptide 1 (GLP-1) signaling through the regulation of specific microRNAs [49]. Previous studies, by simulating physical exertion and hypoxia, have revealed the remarkable effects of HIF-1α: RAB20 is upregulated while TXNIP is downregulated in the tissues examined. This intriguing interplay may shed light on the observed beneficial outcomes in these settings. Collectively, these findings suggest that the stabilization of HIF-1α, especially during muscle contraction and hypoxia, can serve as a formidable ally in countering the onset of insulin resistance, offering a promising avenue for future exploration in metabolic health [47].

### 2.2. Sirtuin 1 (SIRT1)

Sirtuin 1 represents another essential molecular link between OSA and diabetes (Figure 2). Sirtuin 1 levels in blood were found to be decreased in patients with OSA compared to those in the control group. Additionally, 3-month continuous favorable airway pressure treatment restored Sirtuin 1 blood levels and its activity. Studies of type 2 diabetes mellitus (T2DM) in mouse models have shown an increased expression of tyrosine phosphatase 1B, an enzyme that inhibits insulin receptor activity, leading to insulin resistance. Sirtuin 1 inhibits protein tyrosine phosphatase 1B, thereby sensitizing to insulin [50]. The regulatory relationship between hypoxia-inducible factor-1α and SIRT1 is significant. Sirtuin 1 modulates cellular responses to hypoxia by deacetylating HIF-1α. Sirtuin 1 binds to the protein and deacetylates lysine, resulting in the suppression of hypoxia-inducible factor-1α transcriptional activity [45].

### 2.3. MicroRNAs

Circulating microRNA profiles can serve as potential biomarkers for the diagnosis of OSA. Six microRNAs were confirmed to be differentially expressed between non-OSA and OSA patients: microRNA-181a, microRNA-199b, microRNA-345, microRNA-133a, microRNA-340, and microRNA-486-3p. After 6 months of continuous favorable airway pressure treatment, microRNA levels in the OSA group appeared to approach those of the non-OSA group [51]. The most related microRNAs of OSA and diabetes are microRNA-181a and microRNA-199a (Figure 2).

#### 2.3.1. MicroRNA-181a

MicroRNA-181a emerges as a significant biomarker linking OSA to the development of diabetes. Patients with OSA and T2DM exhibited increased expression of microRNA-181a (Figure 2). Moreover, a negative correlation was observed between microRNA-181a and Sirtuin 1 expression, while a positive correlation was noted between microRNA-181a and insulin resistance. This phenomenon may suggest a potential epigenetic pathway for the increased incidence of T2DM in patients with OSA [50].

In kidney tissue affected by diabetes, microRNA-181a specifically targets Kruppel-like factor 6 and early growth response factor-1. These factors play a significant role in the abnormal proliferation of glomerular mesangial cells, the development of tubulointerstitial fibrosis, and increased cell apoptosis. Notably, the downregulation of microRNA-181a observed in patients with obstructive sleep apnea (OSA) suggests that those without OSA may have a greater resilience against the onset of diabetic-related cardiomyopathy or nephropathy [52]. This observation underscores the complex interplay between these conditions and emphasizes the need for further research to deepen our understanding of their relationship.

#### 2.3.2. MicroRNA-199a

MicroRNA-199a represents another important hypoxia-regulated microRNA with implications for the development of diabetes. In hypoxic preconditioning, the downregulation of microRNA-199a was associated with the upregulation of hypoxia-inducible factor-1α and Sirtuin 1, resulting in adaptation to external stimuli (Figure 2). Besides maintaining metabolic homeostasis, SIRT1 acts as a regulator of hypoxia-inducible factor-1α [52].

Diabetic cataract was associated with downregulation of microRNA-199a and its influence on the specific protein 1 gene. In turn, diabetic retinopathy can partially result from the dysregulation of microRNA-199a, related to vascular endothelial growth factor or fibroblast growth factor 7 signaling [52,53].

### 2.4. Inflammatory Markers

OSA’s intermittent hypoxia triggers oxidative stress and inflammation, which promotes insulin resistance, a key factor in diabetes. Conversely, diabetes-related factors like obesity and autonomic dysfunction can exacerbate OSA, further increasing inflammation and making glucose control difficult. The markers of inflammation involved in the bidirectional relationship between OSA and diabetes are tumor necrosis factor-α (TNF-α), interleukin-6 (IL-6), and C-reactive protein (CRP) (Figure 2).

#### 2.4.1. Tumor Necrosis Factor-α

Tumor necrosis factor-α plays a central role in the inflammatory pathway linking OSA to diabetes [54]. This meta-analysis suggests that the overall risk of type 2 diabetes mellitus is strongly associated with elevated levels of inflammatory cytokines, specifically tumor necrosis factor-α, and low levels of adiponectin (Figure 2). For TNF-α, a relatively lower relationship was observed, with an independently increased risk of T2DM, with a relative risk of 1.16 [95% CI 0.87–1.45] [55] When lean subjects with a body mass index of less than 25 kg/m^2^ and patients with a body mass index of 30–40 kg/m^2^ were compared, there was a 7.5-fold increase in tumor necrosis factor-α secretion from adipose tissue. The TNF-α secretion was inversely related to insulin sensitivity, with a correlation coefficient of −0.42 [56]. Tumor necrosis factor-α and its genetic variants are implicated in the development of T2DM due to systemic inflammation, dyslipidemia, and insulin resistance [57].

#### 2.4.2. Interleukin-6

Interleukin-6 serves as another important inflammatory mediator. The meta-analysis comprised 16 cohorts, involving a total of 24,929 participants and 4751 cases. Using data from all trials, a strong positive correlation (1.32 [1.14, 1.51]) was observed between basal plasma IL-6 and T2DM [58].

In contrast to TNF-α, plasma rather than adipose IL-6 demonstrated the strongest relationship with obesity and insulin resistance. Seric IL-6 level was significantly higher in patients with obesity and showed a highly significant inverse relationship with insulin sensitivity, with a correlation coefficient of −0.71 [59].

#### 2.4.3. C-Reactive Protein

C-reactive protein represents the most frequently measured inflammatory marker, synthesized in the liver; there is an apparent association between elevated CRP levels and the risk of T2DM (relative risk 1.48 [95% CI 1.26–1.71]) [60]. CRP is a biomarker that is elevated in OSA patients, reflecting the systemic inflammation caused by intermittent hypoxia and sleep fragmentation.

### 2.5. Adipokine Markers

In individuals with OSA and diabetes, adiponectin and omentin-1 levels are often decreased, while leptin, resistin, and chemerin levels are typically increased. However, the relationship can be complex and is usually mediated by factors such as obesity, intermittent hypoxia, and inflammation (Table 1). These alterations may lead to insulin resistance and metabolic dysfunction, making them potential markers for disease severity, risk stratification, and therapeutic monitoring [61].

#### 2.5.1. Adiponectin

Adiponectin (APN) is a protective adipokine with anti-diabetic properties. Adiponectin is almost exclusively secreted by adipose tissue and appears to act as a hormone that could downregulate inflammatory responses [72].

APN is a key player in the fight against obesity, exhibiting a negative correlation with excess body weight. It activates essential pathways such as AMP-activated protein kinase (AMPK) and peroxisome proliferator-activated receptors (PPAR-α and PPAR-γ), which work together to help regulate body weight. Additionally, APN enhances anti-inflammatory and antioxidative stress responses through its two primary receptors: AdipoR1, predominantly located in skeletal muscles, and AdipoR2, primarily found in the liver. Research indicates that adiponectin levels significantly decrease in patients with T2DM (Figure 2) [73]. This decline is concerning, given adiponectin’s anti-diabetic, anti-inflammatory, and anti-atherogenic properties. Acting as an insulin sensitizer, it plays an essential role in brain function, particularly within the hypothalamus [74]. Notably, individuals with prediabetes exhibit markedly lower adiponectin levels compared to healthy participants, with a positive correlation observed between adiponectin and inflammatory markers such as TNF-α and IL-6 [75].

The relationship between adiponectin and inflammatory markers is complex [76]. In human adipose tissue, APN induces the release of TNF-α and IL-6 through powerful signaling pathways, including Nuclear Factor-kappa B and Extracellularly Regulated Kinase [73,75].

#### 2.5.2. Leptin

Leptin is an essential hormone produced by white adipose tissue and encoded by the Ob (LEP) gene. Leptin regulates appetite and energy expenditure, primarily through its binding to specific long isoform leptin receptors (LepRb) and its interaction with the hypothalamic nucleus [77]. This hormone also engages in a complex relationship with oxidative stress, inflammation, and diabetes, which provides valuable insights into various health conditions (Figure 2) [64]. In cases of OSA, an increase in plasma leptin levels and the development of leptin resistance can be observed, inhibiting the hormone’s physiological action in regulating body fat. These augmented leptin levels induce fat accumulation, with a negative impact on metabolic health and neurological function [78].

Type 2 diabetes mellitus is strongly associated with elevated levels of IL-6, leptin, CRP, and TNF-α [76]. Significant interaction effects were observed between age, body mass index, and diabetes [79,80,81]. The cytokine/adipokine profiles suggest an association between low-grade inflammation and the quality of glucose control [82]. Immune responses regulate leptin; the acute immune response, accompanied by the release of TNF-α and IL-1β, results in a prompt, short-term increase in plasma leptin levels. However, chronic inflammation and its resultant constitutive upregulation of pro-inflammatory cytokines may lead to leptin suppression [83].

#### 2.5.3. Resistin

This adipokine is associated with inflammation and insulin resistance. Previous studies reported elevated resistin levels in patients with OSA and impaired glucose metabolism [67].

#### 2.5.4. Chemerin

Chemerin is a relatively recently discovered adipokine that functions in leukocyte chemotaxis, playing a role in immune responses and inflammation in injured tissues. Increased chemerin levels in OSA induce insulin resistance and inflammation [84,85].

#### 2.5.5. Omentin-1

In both OSA and T2DM, reduced serum level of omentin-1 is associated with disease severity and insulin resistance. The relationship is complex and requires further research; low omentin-1 levels may serve as a predictive marker for OSA and a contributing factor to metabolic dysfunction in these combined conditions [70,71,86].

### 2.6. Other Biomarkers

#### 2.6.1. Melatonin

Melatonin is a hormone produced by the pineal gland that helps synchronize the circadian rhythm with the external environment. Patients with OSA exhibit a significantly lower plasma concentration of melatonin [87]. Melatonin level diminution leads to a marked glucose intolerance and insulin resistance, because its effects are induced through specific, high-affinity G protein-coupled receptors widely expressed in central and peripheral tissues, including beta pancreatic islet cells [88,89].

#### 2.6.2. Orexin

Orexin A and B are neuropeptides produced in the hypothalamus, responsible for regulating sleep and arousal, feeding, stress, and reward responses [90,91,92]. Their effects are due to the binding of orexin receptors [93]. In OSA and diabetes, plasma levels of orexin A decrease [94,95].

#### 2.6.3. Ghrelin

Ghrelin is mainly secreted by specialized endocrine cells (P/D1 cells) [96]. It is a physiological regulator of insulin release from beta cells, ensuring glucose homeostasis [97]. Ghrelin plasma levels are diminished in OSA [98]. Low plasma ghrelin levels are associated with elevated fasting insulin levels and insulin resistance, which can contribute to the development of type 2 diabetes mellitus [99].

## 3. Management of OSA in T2DM Patients

### 3.1. Gold Standard Treatment

Currently, positive airway pressure (PAP) therapy represents the gold standard treatment for patients with moderate to severe OSA, effectively improving AHI and reducing OSA-related symptoms through mechanical support during sleep. These devices increase pressure at the pharyngeal level, thereby widening the pharyngeal airway and preventing it from collapsing [100]. PAP treatment is recommended for patients with moderate to severe OSA (i.e., AHI ≥ 15) and for those with mild OSA who exhibit symptoms (i.e., excessive daytime sleepiness, non-restorative sleep), have comorbidities, or work in critical jobs such as airline pilots, bus and truck drivers, etc. [101].

Continuous positive airway pressure (CPAP) therapy demonstrates statistically significant AHI reduction with a mean difference of −23.49 events per hour (95% CI: −28.68, −18.50) compared to inactive control [102]. Long-term efficacy data show sustained AHI reduction from baseline levels of 49.2 ± 26.1 events/h to 3.4 ± 5.4 events/h at 10-year follow-up [103]. Meta-analysis performed by Li Et Al. on 41 randomized controlled trials involving 7332 patients reported significant improvements in subjective sleepiness (ESS score reduction: −2.14, *p* < 0.001) and objective alertness measures (MSLT improvement: 1.23 min, *p* < 0.001; MWT improvement: 1.6 min, *p* < 0.001) [104]. A recent prospective study on 170 patients with OSA reported 77.4% of patients achieving good adherence after 6 months, with a median residual AHI of 1.2 events/h [105]. PAP therapy reduces postoperative respiratory complications by 28% and unplanned ICU admissions by 56% in surgical patients with OSA.

A recent study based on the Swedish health registry reported a 26% lower risk of all-cause mortality (HR: 0.74; 95% CI: 0.68–0.82; *p* < 0.001) in patients with T2DM and OSA using CPAP Vs. Patients with T2DM who do not use CPAP therapy (EASD 2025, 15–19 September 2025, Abstract 336, available at https://www.emjreviews.com/diabetes/news/easd-2025-cpap-linked-to-lower-mortality-in-people-with-diabetes/, accessed on 13 September 2025).

Patients who do not tolerate or who fail PAP treatment and have surgically correctable airway obstruction, such as tonsil enlargement or retrognathia, and upper airway surgery, can benefit from a different treatment option.

Additional therapeutic methods for patients with mild forms of OSA include mandibular advancement devices (MADs) and neuromuscular electrical stimulation (NMES) to the mouth, tongue, and upper airway. NMES is an oral device that uses mild electrical currents to stimulate and improve the muscle tone of the glossal muscles [106,107,108]. MADs or hypoglossal nerve stimulation (HNS) can be used in patients who are not candidates for surgery. An HNS device is an implantable device that uses mild electrical impulses to stimulate the hypoglossal nerve, preventing airway obstruction. HNS is becoming increasingly used in OSA management [108]. Management of OSA should also include behavioral modification such as weight loss, no supine sleep position, avoidance of alcohol, and sedating medications.

### 3.2. Current and Emerging Therapeutic Interventions

Several therapeutic interventions currently available for treating OSA in patients with diabetes, offering benefits for both conditions, are displayed in Table 2 [109]. It can be observed that GLP-1RAs and SGLT2 inhibitors are the primary drugs used for this purpose (Table 2).

The GLP-1 receptor plays a crucial role in regulating blood sugar and appetite by responding to the GLP-1 hormone. Its main functions include increasing insulin secretion and decreasing glucagon release to lower blood glucose, slowing gastric emptying to promote satiety, and regulating appetite in the brain. Therefore, this receptor is a target for medications used to treat type 2 diabetes and obesity, as these drugs (GLP-1RAs) mimic GLP-1’s actions (Table 2).

Sodium-glucose cotransporter 2 (SGLT2) is a protein in the kidney’s proximal tubule that reabsorbs glucose from the urine back into the bloodstream, playing a key role in glucose homeostasis. It functions by using the sodium gradient to transport glucose and is responsible for reabsorbing about 90% of the filtered glucose. SGLT2 inhibitors block this transporter, leading to increased glucose excretion in the urine (Table 2).

## 4. Anti-Diabetic Drugs Used in OSA: Molecular Mechanisms and Clinical Evidence

Reducing glycemic variations and HbA1c levels, which are necessary in T2DM, may diminish OSA severity [121]. Therefore, modern anti-diabetic drugs, such as GLP-1RA, SGLT2 inhibitors, DPP-4 inhibitors, and biguanides, have been reported to be effective in achieving optimal metabolic control in OSA patients with diabetes.

### 4.1. GLP-1 Receptor Agonists

GLP-1RAs provide multiple effects in various diabetes comorbidities, including hypertension, obesity, metabolic syndrome, and atherosclerotic cardiovascular diseases that commonly co-occur with OSA [110]. This group has 2 representatives: Liraglutide and Tirzepatide. The FDA approved Tirzepatide (Zepbound) as the first medication for the treatment of moderate to severe OSA in adults with obesity, to be used in combination with a reduced-calorie diet and increased physical activity [111]. It represents a significant milestone in the pharmacotherapy of OSA in patients with diabetes.

#### 4.1.1. Molecular Mechanisms of Action of GLP-1RAs

○Core Molecular Signaling Pathways

GLP-1 receptor agonists (GLP-1RAs) influence glucose metabolism through binding to the GLP-1 receptor (GLP-1R), a G-protein-coupled receptor located in vital tissues (the gastrointestinal tract, pancreas, heart, and brain). The activation of GLP-1R sets off a cascade of essential intracellular pathways. Activating adenylyl cyclase increases cyclic AMP (cAMP) levels, leading to protein kinase A (PKA) activation. This essential pathway not only augments insulin secretion but also inhibits glucagon release, thereby contributing to blood sugar regulation. The role of GLP-1 signaling extends further, as it stimulates protein kinase B (Akt) and phosphatidylinositol-3-kinase (PI3K). Both key players are essential in maintaining insulin gene expression and cell survival. The activation of the mitogen-activated protein kinase (MAPK) pathway complements these actions. This pathway supports the proliferation of beta cells and their differentiation, further enhancing insulin production.

○Weight Loss-Mediated Mechanisms

The primary mechanism involves weight reduction through several pathways, including appetite suppression and reduction in fat in the upper airway.

GLP-1RAsdiminish appetite and food intake by directly acting on the Central Nervous System (CNS). They interact with hypothalamic receptors, thus inducing satiety and reducing food consumption.

Emerging evidence suggests that these agents may reduce OSA severity by decreasing upper airway fat deposition [122]. Obesity is strongly associated with obstructive sleep apnea (OSA) through several well-defined mechanisms: (i) the direct accumulation of fat within the upper airway or its walls, which reduces the size of the respiratory lumen; and (ii) changes in muscle structure and fat deposition in the soft tissues, both of which contribute to increased collapsibility of the upper airway [123].

○Central Nervous System (CNS) and Respiratory Control

GLP-1 receptors are present in the central nervous system, particularly in regions that play a critical role in controlling respiration. Activation of these receptors has the potential to positively influence breathing patterns and improve stability. Preclinical studies provide promising evidence that stimulating GLP-1 receptors may enhance respiratory drive and contribute to more stable breathing patterns, highlighting their potential as a target for further research and therapeutic development [124]. The CNS mechanisms include (i) sympathetic nervous system modulation (central GLP-1 activation promotes a food intake-independent shift in nutrient partitioning toward fat utilization, at the expense of lipid deposition rates through sympathetic pathways) and (ii) hypothalamic effects (GLP-1 binds to its receptors in the hypothalamus, activates specific neurons, including those that promote satiety (POMC/CART neurons) and inhibits those that induce hunger (NPY/AgRP neurons)) [124].

○Upper Airway Muscle Tone Enhancement

GLP-1 receptor agonists (GLP-1RAs) may significantly influence the tone of the upper airway muscles by intricately affecting neuromuscular control. This suggests that GLP-1 receptor agonists could act as powerful allies in preventing upper airway collapse during sleep. By preserving muscle tension and coordination, these agonists may substantially reduce the frequency and severity of apneas and hypopneas [124].

○Anti-inflammatory and Cytoprotective Effects

The activation of these cascades leads to a notable reduction in the release of pro-inflammatory cytokines, including interleukin 6 (IL-6), tumor necrosis factor alpha (TNF-α), interleukin 1 beta (IL-1β), interferon gamma (IFN-γ), interleukin 2 (IL-2), and interleukin 17 (IL-17). This response represents a non-T helper cell type 2 (Th2) cytokine profile, underscoring its role in regulating inflammation. There is also a significant decrease in the so-called non-T helper cells 2 (Th2)-mediated cytokine profile and in the release of adhesion molecules such as vascular cell adhesion protein 1 (VCAM-1), intercellular adhesion molecule 1 (ICAM-1), and E-selectin, further enhancing the system’s capacity to manage inflammatory processes effectively [124].

○Pulmonary-Specific: Mechanisms

GLP-1RAs have been found to possess potential bronchodilator effects, which may be attributed to their ability to activate specific transduction pathways that also contribute to anti-inflammatory responses. More specifically, the stimulation of molecular players as cAMP/PKA, cAMP/guanine nucleotide exchange factor (GEF), and phosphatidylinositol-3 kinase (PI3)/PKC in airway smooth muscle cells results in relaxation of these cells [124].

#### 4.1.2. Clinical Evidence

Two clinical trials (SURMOUNT-OSA Trial 1 and SURMOUNT-OSA Trial 2) underline tirzepatide approval as a first medication for OSA in obese adults [125].

SURMOUNT-OSA Trial 1 is a double-blind, randomized, placebo-controlled trial that included 234 participants with moderate-to-severe OSA and obesity (BMI ≥ 30 kg/m^2^) who were not on PAP therapy (were unwilling or unable to use PAP therapy and must not have used PAP for at least 4 weeks before screening). The primary endpoint, change from baseline in apnea-hypopnea index (AHI), was achieved in 52 weeks with a −20.0 events/h (95% CI: −25.8 to −14.2; *p* <0.001) in favor of tirzepatide (−25.3 events/h (95% CI: −29.3 to −21.2)) vs. placebo (−5.3 events/h (95% CI: −9.4 to −1.1)). Secondary endpoints were also confirmed, including AHI percent reduction: 55.0% (tirzepatide) vs. 5.0% (placebo), body weight reduction: 18.1% (tirzepatide) vs. 1.3% (placebo), and disease resolution: 43.0% achieved disease resolution with the highest tirzepatide dose [125].

SURMOUNT-OSA Trial 2 is a double-blind, randomized, placebo-controlled trial that included 235 participants with moderate-to-severe OSA and obesity (BMI ≥ 30 kg/m^2^) who were on established PAP therapy for at least 3 consecutive months before screening and planned to continue PAP therapy during the study. The primary endpoint, change from baseline in apnea-hypopnea index (AHI), was achieved in 52 weeks with a −23.8 events/h (95% CI: −29.6 to −17.9; *p* < 0.001) in favor of tirzepatide (−29.3 events/h (95% CI: −33.2 to −25.4)) vs. placebo (−5.5 events/h (95% CI: −9.9 to −1.2)). Secondary endpoints were also confirmed with AHI percent reduction: 62.8% (tirzepatide) vs. 6.4% (placebo), body weight reduction: 20.3% (tirzepatide) vs. 2.3% (placebo), and disease resolution: 51.5%achieved disease resolution with the highest tirzepatide dose [125].

These studies also evaluated the inflammation pathway, and the reduction in hsCRP (at 52 weeks) was −0.89 mg/dL (95% CI: −1.25 to −0.54; *p* < 0.00001). The magnitude of AHI improvement exceeded the predicted benefit of weight loss alone, suggesting direct effects on the respiratory control center and activation of the anti-inflammatory pathway. The reduction in hsCRP particularly supports the hypothesis that NLRP3 inflammasome modulation might represent a central therapeutic mechanism [125].

A combination of GLP-1 agonists with CPAP therapy demonstrates promising molecular synergies, as evidenced in the SURMOUNT-OSA Trial 2. The mechanical airway stabilization provided by CPAP may enhance the efficacy of GLP-1 agonists by reducing intermittent hypoxia and allowing anti-inflammatory pathways to achieve their maximal therapeutic effect. This combination approach may be particularly beneficial in diabetic patients, where oxidative stress and inflammatory activation can aggravate respiratory impairment [125]. FDA approval was limited to moderate-to-severe OSA in patients with obesity, as the studies were performed, highlighting gaps in our understanding of molecular mechanisms that generate the expected result. This limitation highlights the need for the development of molecular biomarkers to expand therapeutic indications and optimize patient selection strategies [125].

Another molecule that was studied but never received an indication for the treatment of OSA was Liraglutide. Trials using this molecule have proven beneficial in reducing AHI in patients with moderate to severe OSA.

Jiang et al. enrolled 90 patients with T2DM and severe OSA in a 3-month, randomized, controlled, non-blinded clinical trial [126]. The patients were randomized into the liraglutide group that received once-daily liraglutide injections on top of CPAP therapy for OSA and the control group that underwent only CPAP treatment for OSA. The maximum prescribed dose of liraglutide was 1.8mg/dose. The change from baseline in apnea-hypopnea index (AHI) achieved in 3 months was −6.4 events/h, favoring liraglutide (−4.9 events/h) over control (+1.5 events/h). Additionally, BMI decreased significantly by 2 kg/m^2^ and SBP decreased by 4.4 mmHg in the liraglutide group compared to the control group [126].

A randomized, double-blind trial conducted by Blackman et al. included 359 non-diabetic participants with obesity who had moderate or severe OSA and were unwilling/unable to use continuous positive airway pressure therapy [112]. They were randomized for 32 weeks to receive either liraglutide 3.0 mg (180 patients) or placebo (179 patients), both as adjuncts to diet and exercise. The primary endpoint, change from baseline in apnea-hypopnea index (AHI), was achieved with a −6.1 events/h (95% CI, −11.0 to −1.2, *p* = 0.015) in favor of tirzepatide (−12.2 events/h) vs. placebo (−6.1 events/h) [112]. Principal studies using GLP-1 receptor analogs are summarized in Table 3.

### 4.2. SGLT2 Inhibitors (SGLT2i)

#### 4.2.1. Molecular Mechanisms of Action of SGLT2i

○Core Molecular Signaling Pathways

SGLT2 inhibitors are a class of drugs including dapagliflozin, empagliflozin, and ertugliflozin, used as antihyperglycemic agents because they inhibit the sodium-glucose cotransporter 2 (SGLT2) and prevent the reabsorption of filtered glucose from the renal tubular lumen. SGLT2 inhibitors offer a promising therapeutic approach for OSA through multiple molecular mechanisms that target key pathophysiological processes underlying the disease [131].

○Weight loss mechanisms

SGLT2 inhibitors cause glucose excretion, leading to calorie loss (240–320 calories/day), which results in weight reduction as fatty acids are released from fat stores. Clinical studies consistently show a weight loss of 2–4 kg. This is especially important because obesity has been linked to OSA through the mechanisms mentioned earlier. Patients with OSA may experience greater weight loss with SGLT2 inhibition, as both subcutaneous and visceral fat decrease [132].

○Cardiovascular risk reduction mechanisms

Numerous studies have proven the benefits of SGLT2i in reducing heart failure hospitalizations and CV death in patients with and without T2DM [133,134,135].

They primarily act by reducing circulating plasma volume through osmotic and natriuretic diuresis early on and later by suppressing sympathetic nerve activity over the long term. Additionally, SGLT2 inhibitors may lower the incidence of arrhythmias and cardiovascular risk by decreasing oxygen demand and fibrosis. As SGLT2 inhibitors activate sirtuin 1 (and thus PGC-1α and FGF21), they are more cardioprotective than other diabetes medications. These medications play a vital role in managing nighttime hypertension by addressing several key physiological mechanisms. They help reduce circadian sympathetic nerve activity, which is associated with an increased cardiovascular risk during the night. By preventing activation of the sympathetic nervous system, these treatments can effectively lower arterial tone, thereby significantly decreasing the risk of stroke. Moreover, SGLT2 inhibitors are effective in enhancing endothelial function, which is essential for maintaining healthy blood vessels and overall circulation. They achieve this by reducing oxidative stress and inflammation within the blood vessels, fostering a healthier cardiovascular environment. Additionally, these inhibitors target metabolic dysfunction by decreasing insulin resistance and lowering liver enzyme levels, leading to improved metabolic health and a reduced likelihood of related complications. Collectively, these effects highlight the comprehensive benefits of SGLT2 inhibitors in OSA via multiple tissue-protective mechanisms. Other mechanisms include preventing inflammation, enhancing cardiac energy metabolism, inhibiting the sympathetic nervous system, and reducing epicardial fat mass [131,136,137,138,139,140,141,142,143,144,145,146,147,148,149,150,151,152,153,154,155]. A comparison of the origin, mechanistic, and pharmacological effects of both classes of the above-mentioned antidiabetic drugs (GLP-1RAs and SGLT2 Inhibitors), primarily used in the treatment of OSA in T2DM patients, is presented in Table 4.

#### 4.2.2. Clinical Evidence

Throughout their mechanisms of action, SGLT2 inhibitors have been considered a promising perspective for treating OSA in patients with T2DM; therefore, numerous studies have been conducted in this area. Still, no molecule has yet been approved as a treatment for OSA.

One meta-analysis that included nine large randomized controlled trials of SGLT2 inhibitors found that, compared to placebo, SGLT2 inhibitors significantly reduced the incidence of overall respiratory disorders (RR 0.75, 95% CI 0.62–0.91), acute pulmonary edema (RR 0.51, 95% CI 0.29–0.88), asthma (RR 0.57, 95% CI 0.33–0.995), and OSA (RR 0.35, 95% CI 0.12–0.99). Additionally, SGLT2 inhibitors decreased the risks of chronic obstructive pulmonary disease (RR 0.79, 95% CI 0.61–1.02; *p* = 0.073) and pulmonary hypertension (RR 0.43, 95% CI 0.16–1.17; *p* = 0.098). The study concluded that these effects were consistent across various underlying diseases (Psubgroup ≥ 0.209) and different SGLT2 inhibitors (Psubgroup ≥ 0.192) [150]. We have data from both randomized trials and analyses of large RCTs or CVOTs using SGLT2 inhibitors. Studies assessing the relationship between different types of SGLT2 regimens and their effect on AHI in patients with type 2 diabetes mellitus (T2DM) are shown in Table 5.

### 4.3. Metformin

Metformin is the first-line drug used in treating T2DM, as the American Diabetes Association recommends in Standards of Care in Diabetes (available at https://professional.diabetes.org/standards-of-care, accessed on 26 July 2025) [162].

#### 4.3.1. Molecular Mechanisms of Action of Metformin

Metformin enhances insulin sensitivity by increasing the translocation of glucose transporter 4 (GLUT4), thereby improving insulin-stimulated glucose uptake and facilitating muscle glycogen synthesis. This process also activates muscle AMPK, reduces lipolysis and the release of free fatty acids, and increases glucose uptake in skeletal muscle and the liver [163]. It also regulates blood sugar levels by suppressing insulin production in the liver. Metformin inhibits complex I and the α-glycerophosphate shuttle, and the resultant increase in the cytoplasmic NADH/NAD+ ratio diverts glucose precursors away from gluconeogenesis [164]. It also inhibits key gluconeogenic enzymes and reduces the conversion of amino acids and lactate to glucose [165]. Its primary mechanism of action relies on the AMPK activation pathway, which suppresses gluconeogenesis and fatty acid synthesis by inhibiting mitochondrial complex I, reducing ATP production, activating AMPK, and inactivating acetyl-CoA carboxylase [166,167,168,169,170,171,172]. Unlike GLP-1 agonists, metformin’s effects on OSA appear to be primarily mediated through improvements in insulin sensitivity, enhanced mitochondrial function, and reduced oxidative stress [173].

#### 4.3.2. Clinical Evidence

A recent study conducted by Zunica et al. investigated the effect of metformin on glucose metabolism and mitochondrial function in skeletal muscle in a group of 16 patients with obesity and moderate-severe OSA with CPAP treatment. Patients were randomized to receive metformin 2000 mg daily or placebo for 3 months, alongside CPAP therapy. The results showed that metformin improved acute-phase insulin sensitivity and skeletal muscle mitochondrial function and prevented the decline in skeletal muscle respiratory function [174].

Another clinical study, conducted by Lin D. et al., investigated the relationship between metformin therapy and the prevalence of OSA in adult patients with T2DM. This retrospective secondary database analysis included 9853 patients with a one-year follow-up. The study primarily explored the relationship between metformin usage and the prevalence of OSA, a condition that poses significant health risks. In addition to metformin, the researchers examined several other factors, including age, gender, race, the presence of hypertension, congestive heart failure, HbA1c levels (a marker of long-term blood glucose control), and body mass index (BMI). The findings indicated no significant association between metformin use and the likelihood of developing OSA, with an Odds Ratio of 1.17, a Confidence Interval ranging from 1.00 to 1.36, and a *p*-value of 0.049, suggesting borderline significance. Interestingly, the data hinted at a trend indicating an increased prevalence of OSA among those using metformin, warranting further research into this potential connection. Additionally, the study found that lower HbA1c levels were significantly associated with a reduced prevalence of OSA, as evidenced by a *p*-value of less than 0.001. This underscores the importance of glycemic control in influencing health outcomes. In conclusion, while metformin therapy may enhance some aspects of sleep quality, it does not seem to reduce the likelihood of developing OSA [175].

Another study, conducted on 387 patients (mean age 58.4 ± 10.8 years) from the Amiens University Hospital database, compared 314 metformin-treated patients with 73 untreated patients to investigate the relationship between metformin therapy and sleep quantity and quality in patients with T2DM referred for potential sleep disorders. All study participants were subjected to a standardized polysomnographic procedure, forming a robust foundation for the research. A thorough multivariate analysis was conducted, accounting for important factors such as age, gender, BMI, neck circumference, cumulative risk factors, and insulin use. The results revealed that patients treated with metformin experienced significantly longer total sleep time and improved sleep efficiency compared to those not receiving metformin. Specifically, the metformin group reported an impressive total sleep time of 6 h and 39 min, in contrast to only 6 h and 3 min for non-users (*p* = 0.002). Furthermore, their sleep efficiency was considerably higher, recorded at 77.9% ± 12.3 compared to 71.5% ± 17.2 for those not on metformin (*p* = 0.003). These significant differences persisted even after adjusting for various covariates, reaffirming the strength of the association. It is noteworthy that metformin users had a higher average BMI, with medians of 37.5 kg/m^2^ versus 34.8 kg/m^2^ for non-users (*p* = 0.045). In conclusion, the study compellingly demonstrates that metformin therapy is linked to longer sleep duration and enhanced sleep efficiency [176].

For patients with both diabetes and OSA, metformin offers the distinct advantage of being weight-neutral or even promoting weight loss, rather than contributing to weight gain that could worsen OSA symptoms. Therefore, the evidence is weak and calls for the design of better prospective studies.

## 5. Future Therapeutic Research Directions

Future research should prioritize the development of personalized treatment algorithms based on individual molecular biomarker profiles, particularly HIF-1α, inflammatory markers, and adipokine patterns, to optimize therapeutic selection for patients with OSA and diabetes.

Long-term safety and efficacy studies of combination therapies, particularly GLP-1 receptor agonists with SGLT2 Inhibitors and CPAP therapy, are necessary to establish standardized treatment protocols. Additionally, investigation into novel therapeutic targets, such as chronotherapy agents (melatonin receptor modulators and circadian rhythm stabilizers) and dual-mechanism compounds (e.g., aroxybutynin–atomoxetine combinations), represents promising options for addressing the complex pathophysiology of OSA and diabetes comorbidity. Research should also expand beyond obesity-related OSA to include lean patients with diabetes while developing molecular biomarkers as predictive tools for treatment response and monitoring disease progression (Table 6).

## 6. Materials and Methods

A comprehensive literature search was conducted across multiple electronic databases, including PubMed/MEDLINE, Embase, Cochrane Library, and Web of Science, from January 2000 to July 2025. The search strategy employed a combination of terms and keywords related to “obstructive sleep apnea,” “diabetes mellitus,” “molecular biomarkers,” “HIF-1α,” “adipokines,” “inflammatory markers,” “GLP-1 receptor agonists,” “SGLT2 inhibitors,” and “therapeutic interventions.” Two independent reviewers screened titles and abstracts, with full-text articles retrieved for potentially eligible studies. Reference lists of included studies and relevant systematic reviews were manually searched to identify additional studies.

Due to the heterogeneity in study designs, populations, interventions, and outcomes, a narrative synthesis approach was employed rather than quantitative meta-analysis. The synthesis was organized thematically around:○Molecular mechanisms: Grouped by biomarker categories (transcription factors, inflammatory markers, adipokines, hormonal factors)○Therapeutic interventions: Organized by drug class with emphasis on mechanism of action and clinical evidence○Clinical implications: Integration of molecular insights with therapeutic potential

For studies reporting quantitative outcomes, effect sizes were extracted and presented with 95% confidence intervals where available. Clinical significance was interpreted considering the magnitude of biomarker changes, clinically meaningful improvements in AHI (≥15 events/h reduction), and improvements in metabolic parameters.

## 7. Essential Considerations

For patients presenting with the triad of OSA, obesity, and diabetes (or those at risk for diabetes), GLP-1 receptor agonists have proven benefits and should be considered as adjunctive therapy to the conventional one, mainly in patients with obesity or T2DM. SGLT2 Inhibitors represent promising pharmacological candidates for OSA treatment, particularly given their multifaceted benefits beyond glycemic control.

A combined therapy approach (for example, CPAP therapy with pharmacological weight management or an association of GLP-1RAs and SGLT2i) yields superior outcomes compared to monotherapy with either intervention alone. An approach targeting multiple pathways (weight reduction, insulin sensitivity, inflammation, and cardiovascular risk) represents the future of OSA management in patients with diabetes. Additional studies are needed to better explore the mechanisms and outcomes in this population.

### Limitations

This narrative review is subject to several limitations:○Possible selection bias in the literature identification despite a comprehensive search strategy;○Lack of formal statistical analysis due to study heterogeneity;○Potential publication bias favoring positive results;○Heterogeneity in OSA diagnostic criteria and severity classification across studies;○Limited long-term follow-up data for many interventions.

Despite all limitations, this review draws on data from fundamental research and clinical therapeutics to help readers understand the relationship between OSA and diabetes, which evolves from a simple comorbidity to a complex, bidirectional pathophysiological process that can be targeted for intervention. Combining traditional respiratory therapies with modern antidiabetic medications provides new opportunities for comprehensive patient care.

The most recently identified molecular biomarkers and mechanisms—particularly HIF-1α regulation, inflammatory networks, and adipokine dysregulation—have deepened our understanding of the bilateral interconnection between OSA and T1DM. Emerging clinical evidence suggests that GLP-1RAs and SGLT2 inhibitors may reduce the severity of OSA and alleviate daytime sleepiness, potentially mitigating the adverse cardiovascular effects associated with OSA. Moreover, the outcomes of GLP-1 receptor agonist trials in managing OSA support the potential for pharmacological treatments that target multiple pathophysiological pathways simultaneously. Consequently, better glucose control in individuals with diabetes increases HIF-1α protein levels and confers a wide range of benefits, some of which are partly mediated by HIF-1α. All data presented suggest that the severity and impact of untreated OSA may be associated with poor glucose control in type 2 diabetic patients, bolstering the hypothesis that reducing OSA severity could serve as an additional therapeutic strategy.

Finally, the molecular insights extensively discussed in this review provide the foundation for future directions, including new therapeutic strategies that target both respiratory dysfunction and metabolic dysregulation in patients with OSA and diabetes.

## Figures and Tables

**Figure 1 ijms-26-10234-f001:**
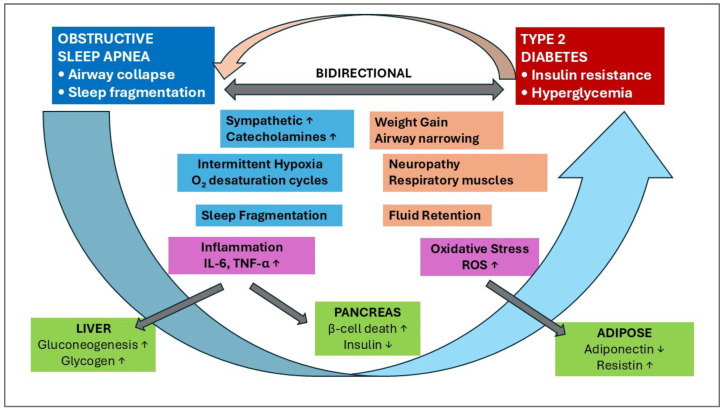
Bidirectional pathophysiology: Obstructive sleep apnea – Type 2 diabetes mellitus; TNF-α = Tumor necrosis factor α; IL-6 = Interleukin 6; CRP = C-reactive protein; RR = relative risk for T2DM; ↑ = Upregulated in OSA; ↓ = Downregulated in OSA; 
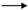
 = Activation.

**Figure 2 ijms-26-10234-f002:**
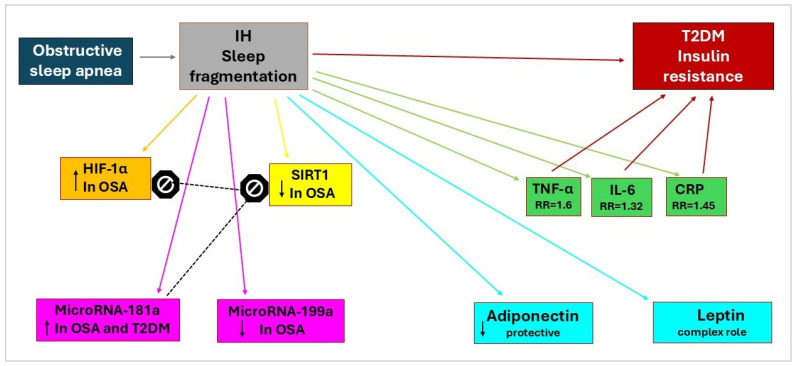
Molecular markers that link OSA with T2DM. OSA = obstructive sleep apnea; T2DM = Type 2 Diabetes; IH = intermittent hypoxia; HIF-1α = Hypoxia-inducible factor 1α; SIRT1 = Sirtuin 1; TNF-α = Tumor necrosis factor α; IL-6 = Interleukin 6; CRP = C-reactive protein; RR = relative risk for T2DM; ↑ = Upregulated in OSA; ↓ = Downregulated in OSA; 
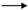
 = Activation; 
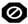
 = Inhibition.

**Figure 3 ijms-26-10234-f003:**
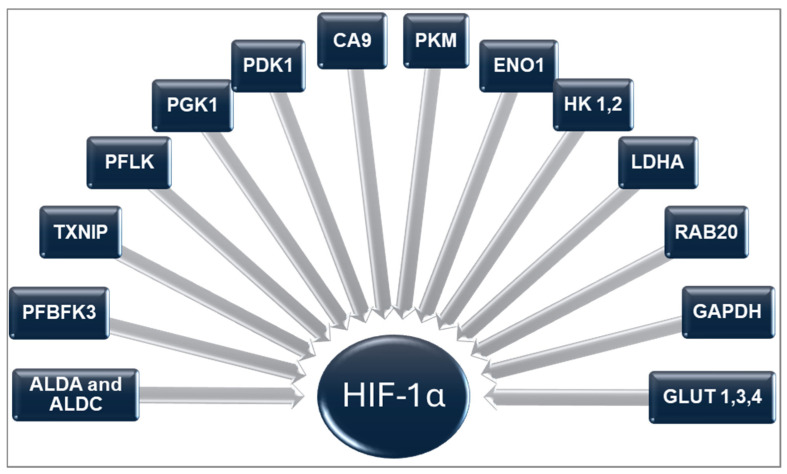
HIF-1 regulated genes: ALDA and ALDC = Aldolases A and C, ENO1 = Eldolase-1 [30], PKM = Pyruvate kinase M, PFKL = Phosphofructokinase L, PGK1 = Phosphoglycerate kinase 1 [37], PFBFK3 = 6-phosphofructo-2-kinase/fructose-2,6- bisphosphate-3 [38], RAB20 = Ras-Related Protein Rab-20 [39], TXNIP = Thioredoxin interacting protein [40], PDK1 = Pyruvate dehydrogenase kinase 1 [41], HK 1,2 = Hexokinases 1 and 2 [42], GAPDH = Glyceraldehyde phosphate dehydrogenase [43], CA9 Carbonic anhydrase-9 [44], GLUT = Facilitative glucose transporter [45].

**Table 1 ijms-26-10234-t001:** Adipokine markers of OSA and T2DM.

Adipokine	Sources	Receptor	Actions	Reference
Adiponectin	Adipocyte	AdipoR1 and AdipoR2; T-Cadherin	Increases insulin sensitivity; Anti-inflammatory	[62,63]
Leptin	White adipose tissue (Obesity gene encoding)	Leptin receptor, (LepR or ObR)	Increases energy consumption; Inhibits fat synthesis; Induces fat decomposition; Inhibits insulin synthesis and secretion.	[64,65]
Resistin	Adipose tissue; Immune and epithelial cells	No mention	Inhibits insulin’s ability to stimulate glucose cellular uptake; Pro-inflammatory	[66,67]
Chemerin	Adipose tissue	Specific receptor proteins: ChemR23 (CMKLR1), and RARRES2	Acts in an immune response; Anti-inflammatory; Regulates glucose metabolism.	[68]
Omentin-1	Omental adipose tissue	No mention	Anti-inflammatory; Regulates fat metabolism; Improves insulin sensitivity.	[69,70,71]

T-Cadherin = Glycosyl-phosphatidylinositol-anchored cadherin; ChemR23 = Chemerin receptor 23 is a G protein-coupled receptor; CMCLR1 = Chemokine-like receptor 1; Chemokine-like receptor 1 is also known as ChemR23; RARRES2 = Retinoic acid receptor responder protein 2; LepR = Leptin receptor, ObR = Obesity receptor; Leptin receptor is also known as Obesity receptor.

**Table 2 ijms-26-10234-t002:** Current and emerging therapeutic interventions for OSA in T2DM patients.

Therapy Class	Specific Agent	Mechanism	OSA Benefits	T2DM Benefits	Clinical Evidence	FDA Status	Key Studies
GLP-1 Receptor Agonists	Tirzepatide	Dual GIP/GLP-1 agonist	52-week study: significant AHI reduction	Superior glycemic control and weight loss vs. GLP-1 alone	Phase III positive results	Approved for obesity, T2DM, and OSA	[110]
	Semaglutide	GLP-1 receptor agonist;	AHI reduction, improved sleep quality, and weight loss	Established weight and glycemic; cardiovascular protection	Phase III trials completed	Approved for obesity, T2DM	[111]
	Liraglutide	GLP-1 receptor agonist	AHI improvement	Established weight and glycemic benefit	RCT evidence	Approved for obesity, T2DM	[112]
SGLT2 Inhibitors	Empagliflozin	SGLT2 inhibition; natriuretic effects	65% OSA occurrence reduction; improved SpO2	Cardiovascular protection; renal benefits, Glycemic control	Meta-analysis evidence	Approved for T2DM, HF, CKD	[113]
	Dapagliflozin	SGLT2 inhibition; natriuretic effects	Reduced fluid retention; improved AHI	Cardiovascular protection; renal benefits, Glycemic control	Observational studies	Approved for T2DM, HF, CKD	[114]
Chronotherapy	Melatonin	Circadian synchronization; antioxidant effects	Sleep architecture improvement	Insulin sensitivity enhancement	Preclinical evidence	OTC supplement	[115]
	Light Therapy	Circadian entrainment; PER2 enhancement	Potential sleep quality improvement	Metabolic rhythm restoration	Early studies	Nonpharmacological	[116]
	CRY Stabilizers	Clock gene stabilization (TW68)	Potential circadian restoration	Hepatic glucose suppression	Preclinical only	Investigational	[117]
Combinations	GLP-1 + SGLT2	Synergistic metabolic effects	Additive OSA benefits potential	Enhanced glycemic/CV outcomes	Ongoing trials	Individual approvals	[118]
	CPAP + GLP-1 /SGLT2	Mechanical + metabolic intervention	Optimal AHI reduction + weight loss	Comprehensive metabolic control	Limited studies	Standard + approved	[119]
	Aroxybutynin +atomoxetine	A selective norepinephrine reuptake inhibitor and a selective antimuscarinic	Activation of the upper airway dilator muscles	No mention	Phase III trials are ongoing	Submitted for approval in OSA	[120]

**Table 3 ijms-26-10234-t003:** Relevant clinical studies using GLP-1RAs in the OSA therapeutic approach.

Reference	Primary and Secondary Objectives	Population/Participants	Sample Size	Intervention/Exposure	Outcome Measures	Major Findings
Malhotra A. et al., 2024 [111]	Primary: To evaluate the change in AHI from baseline. Secondary: To assess percent change in AHI, body weight, hypoxic burden, patient-reported sleep impairment and disturbance (PROMIS scales), hsCRP concentration, and SBP.	Adults with moderate-to-severe OSA (AHI ≥ 15 events/h) and obesity (BMI ≥ 30)	469 (Trial 1: 234 [no PAP], Trial 2: 235 [with PAP])	Tirzepatide (maximum tolerated dose of 10 mg or 15 mg subcutaneously once weekly) vs. placebo for 52 weeks	Change in AHI, percent change in AHI, percent change in body weight, hypoxic burden, PROMIS-SRI and PROMIS-SD scores, hsCRP concentration, and SBP.	In Trial 1, tirzepatide reduced AHI by −25.3 events/h vs. −5.3 with placebo (difference −20.0, *p* < 0.001); body weight by −17.7% vs. −1.6%. In Trial 2, AHI was reduced by −29.3 vs. −5.5 (difference −23.8, *p* < 0.001); body weight by −19.6% vs. −2.3%. Significant improvements in hypoxic burden, PROMIS scores, hsCRP, and SBP
Jiang W. et al., 2023 [126]	Primary: To assess liraglutide’s effect on OSA severity in patients with T2DM. Secondary: To evaluate glycemic control, body weight, and safety.	Patients with T2DM and severe OSA	60	Liraglutide (1.8 mg/day) vs. control	AHI, HbA1c, body weight, adverse events	Liraglutide reduced AHI by 12.2 events/h (*p* < 0.001), improved HbA1c, and reduced body weight, with a tolerable safety profile.
Blackman A. et al., 2016 [112]	Primary: To evaluate liraglutide’s effect on OSA severity in obese individuals. Secondary: To assess changes in body weight and cardio-metabolic outcomes.	Individuals with obesity and with moderate-to-severe OSA	359	Liraglutide (3.0 mg/day) vs. placebo	AHI, body weight, HbA1c, blood pressure	Liraglutide reduced AHI by 12.2 events/h (*p* = 0.015), body weight by 5.7%, and improved cardiometabolic markers compared to the placebo.
O’Donnell C. et al., 2024 [127]	Primary: To compare CPAP and liraglutide on early CV disease markers in OSA. Secondary: To assess changes in AHI and metabolic parameters.	Adults with OSA and obesity	30	CPAP vs. liraglutide (3.0 mg/day)	Carotid intima-media thickness, AHI, HbA1c, body weight	CPAP improved cardiovascular markers (*p* = 0.02) more than liraglutide; however, liraglutide reduced AHI and body weight, with no significant cardiovascular benefit.
Sprung et al., 2020 [128]	Primary: To assess liraglutide with or without CPAP on OSA in patients with T2DM. Secondary: To evaluate glycemic control, body weight, and CV risk markers.	Type 2 diabetes patients with OSA	72	Liraglutide, CPAP, or both vs. placebo	AHI, HbA1c, body weight, cardiovascular risk markers	Study protocol: designed to assess the combined effects of liraglutide and CPAP; results not reported in this paper.
Gomez-Peralta F. et al., 2015 [129]	Primary: To investigate liraglutide’s effect on excessive daytime sleepiness in obese type 2 diabetes patients. Secondary: To assess glycemic control and body weight changes.	Obese patients with type 2 diabetes	158	Liraglutide (1.2–1.8 mg/day)	Epworth Sleepiness Scale (ESS), HbA1c, body weight	Liraglutide reduced ESS scores by 2.9 points (*p* < 0.001), improved HbA1c, and decreased body weight, suggesting benefits for daytime sleepiness.
Baser O. et al., 2024 [130]	Primary: To assess the association between AOMs and the incidence of OSA. Secondary: To compare OSA risk between tirzepatide and semaglutide users.	Patients with obesity (AOM cohort: tirzepatide or semaglutide users; non-AOM cohort: no AOM use)	105,402 (AOM: 20,384; non-AOM: 85,018)	Tirzepatide or semaglutide vs. no AOM	Incidence of OSA, hazard ratio of OSA	The AOM cohort had a lower incidence of OSA (3.12%) compared to the non-AOM cohort (12.56%, *p* < 0.001); AOM use reduced the likelihood of OSA by 40% (HR = 0.60, *p* < 0.0001). Additionally, tirzepatide (2.65%) and semaglutide (3.18%) showed no significant difference (*p* = 0.1664).

AHI—Apnea-Hypopnea Index, AOM—Anti-Obesity Medications, BMI—Body Mass Index, CPAP—Continuous Positive Airway Pressure, CV—Cardiovascular, ESS—Epworth Sleepiness Scale, HbA1c—Glycated Hemoglobin A1c, HR—Hazard Ratio, hsCRP—high-sensitivity C-Reactive Protein, OSA—Obstructive Sleep Apnea, PAP—Positive Airway Pressure, SBP—Systolic Blood Pressure, SN—Serial Number, T2DM—Type 2 Diabetes Mellitus.

**Table 4 ijms-26-10234-t004:** The most significant characteristics of GLP-1RAs vs. SGLT2 Inhibitors.

Characteristic	GLP-1RAs	SGLT2 Inhibitors
Weight Loss Mechanism	Central appetite suppression	Peripheral caloric loss
Primary Site	CNS/GI tract	Kidney
Fluid Effects	Minimal	Diuretic
Respiratory Control	Direct CNS effects and indirect	Indirect via metabolic changes
Onset of Action	Rapid (days-weeks)	Gradual (weeks-months)
Dependency	Receptor-mediated	Non-receptor-mediated

**Table 5 ijms-26-10234-t005:** Relevant clinical studies using SGLT2 inhibitors in the OSA therapeutic approach.

Reference	Primary and Secondary Objectives	Population/Participants	Sample Size	Intervention/Exposure	Outcome Measures	Major Findings
Qiu M et al., 2021 [150]	Primary: To assess the association between SGLT2i and noninfectious respiratory disorders. Secondary: To evaluate specific respiratory outcomes.	Patients with T2DM from randomized trials	42,151	SGLT2 inhibitors vs. placebo or other therapies	Incidence of noninfectious respiratory disorders	SGLT2i were not associated with an increased risk of noninfectious respiratory disorders (RR, 0.95; 95% CI, 0.84–1.07), suggesting safety in this context.
Tang Y et al., 2019 [156]	Primary: To evaluate dapagliflozin’s effect on OSA in T2DM. Secondary: To assess changes in glycemic control and body weight.	Patients with T2DM and OSA	24	Dapagliflozin (10 mg/day)	AHI, HbA1c, body weight	Dapagliflozin reduced AHI (*p* = 0.03), improved glycemic control, and decreased body weight, suggesting potential benefits for OSA in individuals with type 2 diabetes.
Armentaro G et al., 2024 [157]	Primary: To assess SGLT2 inhibitors’ effect on OSA parameters in elderly patients. Secondary: To evaluate CV and metabolic outcomes.	Elderly patients with heart failure, T2DM, and OSA	60	SGLT2i	AHI, oxygen saturation, CV events	SGLT2i improved AHI and oxygen saturation (*p* < 0.05), with benefits in cardiovascular and metabolic parameters in elderly patients.
Mir T et al., 2021 [158]	Primary: To investigate the effect of SGLT2i on sleep apnea in T2DM. Secondary: To assess safety and metabolic outcomes.	Patients with T2DM and OSA from randomized trials	NA	SGLT2 inhibitors vs. control	AHI, AE, glycemic control	SGLT2i significantly reduced AHI (*p* < 0.05) and improved glycemic control, indicating a beneficial role in managing sleep apnea.
Kusunoki M et al., 2021 [119]	Primary: To assess SGLT2 inhibitors’ effect on CPAP initiation in patients with T2DM and OSA. Secondary: To evaluate glycemic control and body weight.	Patients with T2DM and OSA	30	SGLT2i	CPAP initiation rate, HbA1c, body weight	SGLT2i reduced the need for CPAP initiation (*p* < 0.05), with improvements in HbA1c and body weight, suggesting benefits in the management of OSA.
Neeland IJ et al., 2020 [113]	Primary: To evaluate empagliflozin’s effect on OSA in T2DM. Secondary: To assess CV and renal outcomes.	Patients with T2DM and CV disease	7020	Empagliflozin vs. placebo	OSA events, CV death, renal outcomes	Empagliflozin reduced OSA events (HR 0.76, 95% CI 0.59–0.98) and improved cardiovascular and renal outcomes, suggesting broader benefits.
Sawada K et al., 2018 [159]	Primary: To investigate the SGLT2i effect on OSA severity in T2DM. Secondary: To assess metabolic and anthropometric changes.	Type 2 diabetes patients with OSA	24	SGLT2 inhibitors	Apnea-hypopnea index (AHI), body mass index, HbA1c	SGLT2 inhibitors significantly reduced AHI (*p* = 0.02) and improved BMI and HbA1c, indicating potential therapeutic benefits for OSA.
Furukawa S et al., 2018 [160]	Primary: To assess dapagliflozin’s effect on sleep-disordered breathing in obese T2DM. Secondary: To evaluate body weight and glycemic control.	Japanese patients with obesity and T2DM	30	Dapagliflozin (5 mg/day)	Apnea-hypopnea index (AHI), body weight, HbA1c	Dapagliflozin reduced AHI (*p* < 0.05), body weight, and HbA1c, demonstrating its efficacy in improving sleep-disordered breathing.
Butt JH et al., 2024 [161]	Primary: To evaluate dapagliflozin’s effect on sleep apnea in heart failure and type 2 diabetes patients. Secondary: To assess CV outcomes.	Heart failure patients with or without T2DM	11,007	Dapagliflozin vs. placebo	Sleep apnea events, heart failure hospitalization, CV death	Dapagliflozin reduced sleep apnea events (HR 0.79, 95% CI 0.64–0.97) and improved heart failure and cardiovascular outcomes.

AE—Adverse Events, AHI—Apnea-Hypopnea Index, BMI—Body Mass Index, CI—Confidence Interval, CPAP—Continuous Positive Airway Pressure, CV—Cardiovascular, HbA1c—Glycated Hemoglobin A1c, HR—Hazard Ratio, NA—Not Available, OSA—Obstructive Sleep Apnea, RR—Risk Ratio, SGLT2i—Sodium-Glucose Cotransporter 2 inhibitors, SN—Serial Number, T2DM—Type 2 Diabetes Mellitus.

**Table 6 ijms-26-10234-t006:** Key molecular biomarkers as therapeutic targets.

Molecular Marker	Clinical Relevance	Therapeutic Target	References
HIF-1α	Increase in OSA patients; correlates with insulin resistance; promotes inflammation.	HIF-1α stabilizers; circadian modulators	[45]
TNF-α	Elevated in OSA; correlates with CIH severity	Anti-TNF therapies; adipokine modulators	[177]
IL-6	Acute phase reactant; hepatic glucose production	JAK inhibitors; IL-6 blockers	[178]
CRP	Correlates with OSA severity and diabetes risk	Anti-inflammatory agents	[179]
Leptin	Resistance in obesity; maintains inflammation despite metabolic dysfunction.	Leptin sensitizers; circadian modulators	[180,181,182]
Adiponectin	Reduced in both OSA and T2DM; protective against metabolic dysfunction	Adiponectin receptor agonists	[78]
Resistin	Elevated in metabolic dysfunction	Adipokine modulators	[67]
ROS/Antioxidants	Activates NF-κB; impairs insulin signaling	Antioxidant supplementation; SOD mimetics	[183,184,185]
miRNA-181a	Altered in OSA; links to insulin resistance	miRNA modulators	[186,187]

## Data Availability

No new data were created or analyzed in this study. Data sharing is not applicable to this article.

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
