# Peer review of "Molecular Biomarkers and Therapeutic Approach of Patients with Diabetes and Obstructive Sleep Apnea"

_ijms, 2025, doi:10.3390/ijms262010234_

Round 1

Reviewer 1 Report

Comments and Suggestions for Authors

Comments and Suggestions for Authors

This manuscript discusses the relationship between T2DM and patients with OSA syndrome from the aspects of key biomarkers (HIF-1α, inflammatory mediators, etc.), drugs and methods of treatment. However, I have some opinions regarding certain aspects of it.

  1. Lines 58-67 Although the first sentence pointed out that the co-occurrence of OSA and diabetes mellitus is remarkably high., the last sentence suddenly described T1DM as a bit abrupt.
  2. Lines 123-125 Delete “Hyperglycemia interferes with the function of HIF-1α.”. Change “hypoxia-inducible factor-1α” to “HIF-1α”. In addition, what is the relationship between proline hydroxylation and HIF-1α?
  3. Lines 128-145 What is the relationship between HIF and HIF-1α? Are they not the same substance?
  4. Figure 3 perhaps these genes should be classified into categories.
  5. Lines 188 Hypoxia-inducible factor-1α has appeared in the full text, so all the abbreviations are used except the full name for the first time. Please check and revise the full text.
  6. Lines 383-386 The main principles of GLP-1 and SGLT2 can be briefly described at the end.

Author Response

C. This manuscript discusses the relationship between T2DM and patients with OSA syndrome from the aspects of key biomarkers (HIF-1α, inflammatory mediators, etc.), drugs, and methods of treatment. However, I have some opinions regarding certain aspects of it.

R. The authors are grateful to Reviewer 1 for the time accorded to review this manuscript and for the valuable comments aimed at improving its quality.

C1. Lines 58-67: Although the first sentence pointed out that the co-occurrence of OSA and diabetes mellitus is remarkably high, the last sentence suddenly described T1DM as a bit abrupt.

R1. The authors thank Reviewer 1 for this accurate comment. They revised this point as follows:

Lines 60-61: The co-occurrence of OSA and both type 1 and type 2 diabetes mellitus is remarkably high.

Lines 65-66: The association between OSA and type 1 diabetes (T1DM) has also been evidenced [10–13], with an almost 51.9% [14].

C2. Lines 123-125: Delete “Hyperglycemia interferes with the function of HIF-1α.”. Change “hypoxia-inducible factor-1α” to “HIF-1α”. In addition, what is the relationship between proline hydroxylation and HIF-1α?

R2. The authors are grateful for Reviewer 1's attention and insightful comments. Reviewer 1 is invited to find the authors’ responses to both aspects below:

R2. Part 1. Deleted.

R2. Part 2. Lines 135-144:

HIF-1α has a low half-life time under normoxic conditions; it has two key proline hydroxylation sites, Pro402 and Pro564, which are both involved in targeting the protein for degradation. Prolyl hydroxylases (PHDs) add hydroxyl groups to specific proline residues on the HIF-1α protein. This hydroxylation creates a recognition site for the von Hippel-Lindau (VHL) tumor suppressor protein. The VHL protein is part of an E3 ubiquitin ligase complex that "marks" HIF-1α for degradation through the ubiquitin-proteasome pathway [32]. Therefore, HIF-1α levels are maintained low, and its activity is suppressed in the presence of sufficient oxygen.

The lack of oxygen inhibits PHDs, and the VHL complex cannot bind to HIF-1α. Therefore, HIF-1α is not marked for degradation and becomes stable.

C3. Lines 128-145: What is the relationship between HIF and HIF-1α? Are they not the same substance?

R3: The authors appreciate Reviewer 1 carefully checking and this attentive comment. They marked their response in lines 126-134:

HIF is a heterodimer composed of two units: an α-subunit, which is oxygen-regulated, and a constitutively expressed β-subunit, belonging to the helix-loop-helix Per/Arnt/Sim transcription factor family [29]. Three analogs of HIF α-subunits are known: HIF-1α, HIF-2α (established regulatory factors), and HIF-3α (with an uncertain role) [30].   The first one, HIF-1α, is the best-examined HIF α-subunit [31]. Although its transcriptional level remains stable, the HIF-1α protein is highly unstable under normoxic conditions, which is attributed to the presence of an oxygen-dependent degradation domain [31]. In low oxygen conditions (hypoxia), this hydroxylation is inhibited, stabilizing HIF-1α and allowing it to activate genes that promote survival.

C4. Figure 3 perhaps these genes should be classified into categories.

R4. The authors are grateful for this valuable comment. All genes are classified in Lines 153-165:

The majority of genes are related to glycolysis enzymes: hexokinases (HK 1,2), which transform glucose into glucose-6-phosphate; liver type phosphofructokinase (PFKL); 6-phosphofructo-2-kinase/fructose-2,6-bisphosphate-3 (PFBFK3); phosphoglycerate kinase 1 (PGK1); aldolases A and C (ALDA and ALDC); glyceraldehyde-3-phosphate dehydrogenase (GAPDH); enolase 1 (ENO1), which transforms 2-phosphoglycerate into phosphoenolpyruvate; pyruvate kinase M (PKM), which is responsible for the fi-nal step of glycolysis. Other genes regulate (i) glucose transport, including glucose transporters (GLUT 1, 3, 4), (ii) lactate metabolism, involving lactate dehydrogenase A (LDHA) and pyruvate dehydrogenase kinase (PDK1), and (iii) maintain pH levels during hypoxia through carbonic anhydrase 9 (CA9). Several genes are associated with other functions: metabolic regulation (thioredoxin-interacting protein, TXNIP) and vesicular trafficking (small GTP-ase, RAB20). All are presented in Figure 3.

C5. Line 188: Hypoxia-inducible factor-1α has appeared in the full text, so all the abbreviations are used except the full name for the first time. Please check and revise the full text.

R5. The authors thank Reviewer 1 for this attentive comment. They checked and corrected this aspect throughout the entire manuscript.

C6. Lines 383-386: The main principles of GLP-1 and SGLT2 can be briefly described at the end.

R6. The authors are grateful for this valuable comment. They responded in lines 410-420 as follows:

The GLP-1 receptor plays a crucial role in regulating blood sugar and appetite by responding to the GLP-1 hormone. Its main functions include increasing insulin secretion and decreasing glucagon release to lower blood glucose, slowing gastric emptying to promote satiety, and regulating appetite in the brain. Therefore, this receptor is a target for medications used to treat type 2 diabetes and obesity, as these drugs (GLP-1RAs) mimic GLP-1's actions (Table 2).

Sodium-glucose cotransporter 2 (SGLT2) is a protein in the kidney's proximal tubule that reabsorbs glucose from the urine back into the bloodstream, playing a key role in glucose homeostasis. It functions by using the sodium gradient to transport glucose and is responsible for reabsorbing about 90% of the filtered glucose. SGLT2 inhibitors block this transporter, leading to increased glucose excretion in the urine (Table 2).

Reviewer 2 Report

Comments and Suggestions for Authors

The review was prepared efficiently and professionally. I have read it with interest and I have no comments. 

The review is devoted to an urgent problem - the relationship between sleep apnea and diabetes mellitus. These conditions continue to be a problem in diagnosis and treatment. The issues of personalizing the treatment of these common concomitant diseases require an interdisciplinary approach. The review identifies the most significant biomarkers that may become promising areas for the development of new treatments for these diseases along with existing approaches. The opportunity to search for new diagnostic and therapeutic approaches using biomarkers is timely and necessary. The relevance of this article is beyond doubt, as these conditions are a common problem in both somnology and endocrinology. The review is an example of a thorough analysis of current and relevant literature data on the subject. Therefore, the issue raised in the article is relevant and timely. The drawings presented in the review serve to emphasize important points and aid in understanding the material. This review was very informative for me as a somnologist and scientific researcher!

Author Response

C. The review was prepared efficiently and professionally. I have read it with interest and I have no comments. 

The review is devoted to an urgent problem - the relationship between sleep apnea and diabetes mellitus. These conditions continue to be a problem in diagnosis and treatment. The issues of personalizing the treatment of these common concomitant diseases require an interdisciplinary approach. The review identifies the most significant biomarkers that may become promising areas for the development of new treatments for these diseases along with existing approaches. The opportunity to search for new diagnostic and therapeutic approaches using biomarkers is timely and necessary. The relevance of this article is beyond doubt, as these conditions are a common problem in both somnology and endocrinology. The review is an example of a thorough analysis of current and relevant literature data on the subject. Therefore, the issue raised in the article is relevant and timely. The drawings presented in the review serve to emphasize important points and aid in understanding the material. This review was very informative for me as a somnologist and scientific researcher!

R. The authors are grateful to Reviewer 2 for the time dedicated to reviewing the present manuscript, the accuracy involved in all mentions, and the excellent appreciation for their hard work.